# Assessing the effectiveness of the community participation approaches to improve access to mass drug administration for trachoma elimination in a pastoral conflict area of Baringo County, Kenya

**Collins Okoyo**[1,2,3*], **Omar Kopi**[1,3], **Paul M. Gichuki**[1], **Bridget W. Kimani**[1], **Tabitha Kanyui**[1], **Titus Waititu**[4], **Wyckliff P. Omondi**[4], **Doris W. Njomo**[1]

1 Eastern and Southern Africa Centre of International Parasite Control, Kenya Medical Research Institute, Nairobi, Kenya, 2 Department of Epidemiology, Statistics and Informatics, Kenya Medical Research Institute, Nairobi, Kenya, 3 Department of Data Management and Analysis, Colozzy Data Analytics and Research Solutions, Nairobi, Kenya, 4 Vector-Borne and Neglected Tropical Diseases Unit, Ministry of Health, Nairobi, Kenya

* comondi@kemri.go.ke

## Abstract

### Background

Trachoma remains a significant public health issue in many regions, including Baringo County, Kenya. Despite the ongoing mass drug administration (MDA) campaigns in Baringo, the achievement of optimal treatment coverage has been hindered by factors such as conflicts and the nomadic lifestyle dominating this region. To address these challenges, innovative strategies are needed to improve community engagement and enhance MDA uptake. This study evaluated the effectiveness of community participation approaches in improving MDA access among residents of Baringo County.

### Methods

The study used a pre- and post-intervention design, utilizing a systematic random sampling of households. The study area was Loyamorok Ward, Tiaty East Sub-County in Baringo County. The county was purposively selected due to its historical challenges in achieving optimal treatment coverage for trachoma, including its nomadic lifestyle, intercounty border movements, and persistent conflicts. A sample of 350 respondents was randomly selected for the pre- and post-intervention surveys. Data were collected using a structured questionnaire, which captured information on socio-demographic and socio-economic characteristics, knowledge about trachoma and MDA, drug use, and perceptions of the treatment. Generalized linear models were employed to estimate the likelihood of MDA access before and after the

**Data availability statement:** All relevant data are included within the paper.

**Funding:** This work received financial support from the United States Agency for International Development (USAID) through its Neglected Tropical Diseases Program through their support of the Coalition for Operational Research on Neglected Tropical Diseases (COR-NTD) grant to DN, through grant number NTDSC 241U. The funders had no role in study design, data collection and analysis, decision to publish, or preparation of the manuscript.

**Competing interests:** The authors have declared that no competing interests exist.

intervention and the impact of the intervention, which incorporated time difference as an interaction term in the models.

## Principal findings

The results indicated a significant increase in community participation and MDA access, with the proportion of participants who took drugs during the last MDA significantly rising from 72.4% pre-intervention to 92.9% post-intervention (Diff=-0.205, z=-5.68, p<0.001). Type of occupation was found to significantly impact access to MDA for trachoma, participants doing pastoral activities (aOR = 11.45, 95% CI: 1.15-114.14, p=0.038), and those who were not engaged in work outside the home (housewife) (aOR = 12.87, 95% CI: 1.09-151.52, p=0.042) showed significant increased access to MDA compared to salaried workers. The study showed that knowledge about trachoma and MDA significantly improved after the implementation of the interventions. Awareness about MDA increased from 45.7% during pre-intervention to 53.3% post-intervention.

## Conclusions

These findings suggest that the implemented community participation strategies positively influenced MDA uptake. The implemented interventions should be considered for wider application to enhance treatment coverage and accelerate trachoma elimination efforts in Kenya.

## Author summary

Trachoma remains a significant public health concern in many rural communities with high rates of poverty and limited access to healthcare. The surgery, antibiotics, facial cleanliness and environmental hygiene (SAFE) strategy was designed by the World Health Organization (WHO) as a means of accelerating trachoma elimination. Delivery of antibiotics through mass drug administration (MDA) is effective in reducing prevalence of active trachoma and prevention of disease transmission. However, achieving adequate treatment coverage during MDA in challenging environments, such as pastoral and conflict-affected regions, remains a challenge. This study aimed to evaluate the effectiveness of a community-participatory approach in improving trachoma treatment coverage in Loyamorok Ward, Baringo County, a region characterized by a nomadic lifestyle, intercounty border movements, and low MDA uptake. The study employed a pre- and post-intervention design, utilizing a systematic random sampling of households to collect quantitative data. Key interventions, including enhanced communication, optimized drug distribution, timely MDA implementation, and capacity building, were implemented during the intervention phase. The results demonstrated a significant increase in MDA uptake and improved awareness of trachoma and its prevention. These findings suggest that the community-participatory approaches can be a valuable tool in addressing the challenges of trachoma elimination in similar settings.

## Introduction

Neglected tropical diseases continue to pose a major public health challenge in many parts of the world, particularly in resource-limited settings [1]. Trachoma is a neglected tropical eye disease caused by infection with the bacterium *Chlamydia Trachomatis* [2]. The infection spreads through direct contact with ocular and nasal secretions, as well as indirectly via fomites (like clothing) and eye-seeking flies [3,4]. Trachoma remains the leading infectious cause of blindness globally [5–7]. Approximately 103 million individuals reside in regions endemic to trachoma, putting them at risk of vision loss [8].

Trachoma predominantly affects impoverished, rural communities in low-income countries, especially in sub-Saharan Africa [9]. It is strongly associated with poverty, poor sanitation, and limited access to water [10]. A four-thronged strategy that includes Surgery for trichiasis, Antibiotics to treat active infection, Facial cleanliness, and Environmental improvement, commonly known as the SAFE strategy, is the cornerstone of trachoma control efforts. The antibiotics component of the strategy recommends three effective rounds of mass drug administration (MDA) in the whole population in communities where prevalence is more than 10% in children aged 1–9 years [11]. An effective treatment coverage of at least 80% is considered successful [12,11].

The Ministry of Health in Kenya spearheads MDA programmes for trachoma elimination, with crucial support from partners such as the Fred Hollows Foundation and Operation Eyesight Universal. Following programme dosing guidelines [13], the aim is to treat all eligible members within targeted communities during each MDA campaign. To determine the reach of these campaigns, MDA coverage in Kenya is calculated by dividing the number of individuals who received the prescribed medication by the total population eligible for treatment within the defined geographical area. The denominator for this calculation is typically derived from the most recent national population census data. Surveys, such as post-MDA coverage surveys, are frequently employed to independently verify the reported administrative treatment numbers and assess the overall accuracy of coverage estimates achieved during the campaign. These surveys provide valuable data for understanding the true reach of the MDA and identifying any gaps in coverage.

Despite the widespread use of MDA with antibiotics as the primary strategy to combat trachoma in Kenya, and the frequent employment of community-based approaches like community meetings, door-to-door health education, and participatory planning, the elimination of trachoma remains a significant challenge, particularly in pastoral and conflict-affected areas. Baringo County, Kenya, exemplifies such a region where trachoma is endemic and the population grapples with ongoing security challenges due to inter-communal conflicts. Therefore, this study aimed to assess the effectiveness of a community participatory approach implemented to improve MDA uptake for trachoma control, specifically within this challenging context. By considering contextual factors, social and environmental influences, and program-level variables, we aimed to identify strategies that can enhance community engagement and ultimately contribute to the successful elimination of trachoma in similar settings.

## Materials and methods

### Ethics statement

Ethical approval for this study was obtained from the Scientific and Ethics Review Unit (SERU) of the Kenya Medical Research Institute (KEMRI/SERU/4532). Data collectors were trained on ethical guidelines for the protection of human research participants and were made aware of the ethical principles governing research involving human subjects. Prior to participation, all participants (who were aged 18 years and above) reviewed written informed consent forms and provided their signed consent to have their information collected. The household heads provided information on behalf of their household members. Data collection was conducted in private settings to ensure participant confidentiality and privacy.

### Study area

Baringo County, located in the heart of Kenya's Rift Valley region, covers an area of 10,976.4 km² [14]. With a population of 666,763 according to the 2019 census, the county is divided into seven administrative sub-counties: Baringo Central, Baringo

North, East Pokot, Eldama Ravine, Baringo South, Mogotio, and Tiaty East. Tiaty East Sub-County, bordering Turkana County to the northeast and Baringo North Sub-County to the south, has a total population of 73,424. Of this population, 38,356 are male, and 35,068 are female [14]. The sub-county comprises of four wards: Silale, Loyamorok, Tangulbei, and Churo-Amaya. This study was conducted in Loyamorok Ward, purposefully selected due to its low treatment coverage trends, nomadic lifestyle, intercounty cross-border movement, and conflicts. Covering an area of 597.8 km², the ward has a population of 13,885 [14]. The primary economic activity in the ward is animal trading, involving cows, goats, sheep, donkeys, and camels.

## Study design and setting

This study employed a quasi-experimental design with a pre-intervention, intervention, and post-intervention phase. Quantitative data collection methods were utilized in the pre- and post-intervention phases. The November 2021 MDA served as the basis of inquiry for the pre-intervention phase to identify groups with consistently low participation and access to MDA and the assessment of the barriers related to contextual, social, environmental, and programmatic factors hindering their engagement. During this phase, feasible and field-applicable opportunities and strategies were identified to reach these groups in the villages of Loyamorok Ward. The identified opportunities and strategies were then developed using community-based participatory approaches, emphasizing community participation and action. The intervention phase involved testing the developed strategies and the identified opportunities in the study villages prior to and during the May 2023 MDA. Subsequently, an impact assessment (post-intervention) was conducted to evaluate the impact of the implemented strategies and the identified opportunities on improving community participation, access to MDA, and treatment coverage for trachoma elimination as a public health concern.

## Sampling and study population

This study included all nine villages in Loyamorok Ward. Each village served as the primary sampling unit, also referred to as the stratum. A systematic random sampling technique was used to select households within each village. The study population included adult participants (18 years and above) who were heads of households, and they responded on behalf of themselves and their household members during the surveys.

## Data collection and management

Participant's responses were captured electronically using open data kit (ODK), a mobile-based data collection system that included inbuilt data quality checks to prevent errors. Data was collected using an interviewer-based questionnaire that was administered to consenting household heads (Supplementary file SI1). The questionnaire captured information on socio-demographic and socio-economic characteristics, knowledge about trachoma and MDA, trachoma drug uptake, and perception of the treatment. Hard copies of the quantitative data were stored in secure, lockable cabinets for backup purposes. Soft copies were stored on password-protected computers and tablets, with authorized access limited to the principal investigator to maintain quality control. Interview data was transmitted to a secure server in Nairobi via mobile network. Data was collected from 30th January to 2nd February of 2023 during pre-intervention survey, and 7th to 13th October of 2023 during post-intervention survey.

## Statistical analysis and modelling

Quantitative data were analyzed using STATA version 16.1 (STATA Corporation, College Station, TX, USA). Participants' responses were pooled and arranged in different categories. All proportions were calculated for variables of interest, and their 95% confidence intervals (CIs) were calculated using generalized linear models that accounted for villages (clusters). Access to trachoma drugs was defined as the proportion of participants who took the trachoma drugs during the last MDA. Likelihood of MDA access was estimated using the generalized linear models, reporting the unadjusted odds ratios

for univariable analysis and adjusted odds ratios for multivariable analysis, and these ratios were compared between the pre-intervention and post-intervention surveys. Minimum adequate variables were selected for multivariable analysis using a pre-specified inclusion criterion of p-value < 0.2 in a forward variable selection method, which selected covariates meeting the set criterion. Further, the intervention impact was similarly estimated using the generalized linear models while using time difference as the interaction term in the models.

## MDA history of the study area

Although surveys conducted between 2004–2011 showed that there was no burden of active trachoma in Baringo County, further investigations conducted in Tiaty East and Tiaty West sub-counties (combined as one evaluation unit (EU)) showed a prevalence of 34.4% active trachoma, amongst the highest in the country. Implementation of MDA in the two sub-counties was therefore initiated in the year 2012 and conducted for five rounds up to 2017, with treatment coverage results as shown in Table 1. In the year 2018, an impact assessment survey was conducted, and the results showed a prevalence of 12.8% active trachoma. According to this impact result, three effective MDA rounds were required. The treatment rounds were conducted in 2020, 2021, and 2023.

Although the treatment coverage achieved in 2020 and 2021 was 95.6% and 79.7%, respectively, further investigation at the lower levels (wards) showed coverages ranging between 48% to 57%, which are far below the recommended threshold. Investigations in Loyamorok ward showed low treatment coverage, ranging from 56.6% and 67.6% respectively in 2020 and 2021. Further investigation of the treatment coverage achieved in six of the nine villages of Loyamorok Ward showed a coverage of 48.1%, 48.6%, 49.3%, 55.7%, 56%, and 57.1% in 2021. Loyamorok ward faces challenges related to drought and hunger, a nomadic lifestyle with movement in search of pasture, coupled with conflicts related to cattle rustling, resulting in insecurity in the area.

The third MDA round (2023 MDA) was used to test the interventions reported in this study. The 2023 MDA achieved a high coverage of 92.3% in the whole EU and 87.0% in Loyamorok Ward specifically.

## The tested interventions

**Effective communication and community engagement.** To effectively communicate information about trachoma and the MDA exercise, a tailored approach was employed. By engaging diverse stakeholders, key messages were crafted and disseminated through locally relevant channels. This ensured that the community was well-informed about the disease, the importance of treatment, and the upcoming MDA.

**Optimized drug distribution.** A multifaceted distribution strategy was implemented to maximize drug accessibility. In addition to door-to-door distribution, which was identified as the preferred method by community members, other

**Table 1. MDA coverage history of the study area, Tiaty East and Tiaty West sub-counties (combined as one evaluation unit).**

| Year | Number targeted | Number treated | Treatment coverage (%) |
|---|---|---|---|
| 2012 | 148,097 | 91,049 | 61.5 |
| 2013 | 149,905 | 116,011 | 77.4 |
| 2014 | 154,402 | 101,067 | 65.5 |
| 2015 | 155,365 | 127,682 | 82.2 |
| 2017 | 173,182 | 104,911 | 61.0 |
| 2020 | 153,347 | 146,673 | 95.6 |
| 2021 | 163,381 | 130,266 | 79.7 |
| 2023 | 174,043 | 160,625 | 92.3 |

approaches, such as centralized distribution at health facilities, and utilization of community leaders and health workers, were employed. This flexible approach aimed to cater for the diverse needs and preferences of the community.

**Timely MDA implementation.** To optimize the timing of the MDA, the rainy season was selected based on community input. This period was considered the most suitable time due to the high number of community members being available in their homes. Careful planning and execution were crucial in ensuring the successful implementation of the MDA campaign during this optimal window.

**Enhanced capacity building.** To enhance the effectiveness of the MDA teams, capacity-building initiatives were undertaken. The capacity-building initiatives included training, seminars, and workshops, among others. These initiatives addressed specific concerns raised by community members during the pre-intervention phase, such as inadequate drug distribution, poor communication, and insufficient drug supply. By empowering the drug distributors with knowledge and skills, the teams were better equipped to deliver the MDA efficiently and effectively.

## Results

During the pre- and post-intervention surveys, nine villages were surveyed from Loyamorok Ward in Tiaty East Sub-County, Baringo County (Table 2). The study villages were selected prior to the start of the surveys based on the reported low coverage of MDA in the county and following stakeholder engagements.

### Socio-demographic characteristics of the participants

Overall, 350 respondents participated in the pre-intervention study approximately 100 days prior to the start of MDA. Almost a similar number (351) of respondents participated in the post-intervention survey, approximately 120 days after the MDA. Information on age was obtained from all the participants with a median age of 36 years (interquartile range (IQR): 22 years) during the pre-intervention survey and 36 years (IQR: 21 years) during the post-intervention survey. Similarly, gender information was collected from all the participants with male being 136 (38.9%) and female 214 (61.1%) during pre-intervention, and male 166 (47.3%) and female 185 (52.7%) during post-intervention (Table 3).

Of all the respondents surveyed during both survey time points, 261 (74.6%) and 251 (71.5%) were currently married during pre-intervention and post-intervention respectively, and 202 (57.7%) and 206 (58.7%) were Christians during pre-intervention and post-intervention respectively.

**Table 2. Number of households selected in each village during pre- and post-intervention surveys in Loyamorok Ward, Tiaty East, Baringo County.**

| Ward | Village | Number of households | |
|---|---|---|---|
| | | Pre-intervention | Post-intervention |
| Loyamorok | Chepungus | 41 | 17 |
| | Atirirai | 48 | 56 |
| | Nyaunyau | 24 | 68 |
| | Chesiran | 40 | 20 |
| | Chesesoi | 53 | 34 |
| | Angorok | 43 | 37 |
| | Cheptaran | 38 | 34 |
| | Cherelio | 22 | 33 |
| | Loyamorok | 41 | 52 |
| Total | | 350 | 351 |

Table 4 assessed the association between selected socio-demographic factors and access to trachoma drugs using univariable analysis. Participants who were currently married showed significantly increased access to MDA during post-intervention compared to those who were single/divorced/widowed (OR = 2.16, 95% CI: 1.32-3.54, p = 0.002). Additionally, participants aged between 20–30 years showed significantly increased access to MDA during post-intervention compared to those aged 60 years and above (OR = 2.44, 95% CI: 1.03-5.79, p = 0.043). Regarding religion, Christian participants (OR = 24.15, 95% CI: 2.83-206.15, p = 0.004) and non-practising participants (OR = 9.05, 95% CI: 1.06-77.29, p = 0.044) showed significantly increased access to MDA during post-intervention compared to those in other religions.

Table 5 assessed the association between selected socio-demographic factors and access to trachoma drugs using multivariable analysis. Participants who were currently married showed significantly increased access to MDA during post-intervention compared to those who were single/divorced/widowed (aOR = 2.01, 95% CI: 1.21-3.36, p = 0.007). Regarding religion, Christian participants (aOR = 24.81, 95% CI: 2.87-214.27, p = 0.004) and non-practising participants (aOR = 9.93, 95% CI: 1.15-85.87, p = 0.037) showed significantly increased access to MDA during post-intervention compared to those in other religions. Assessment of the intervention impact (changes in odds ratio pre- and post-intervention) indicated that no socio-demographic factor was significantly associated with the increased access to MDA.

## Socio-economic characteristics of the participants

Table 6 shows selected socio-economic factors which included education level, occupation, presence of toilet, toilet type, type of roofing, flooring, and walling, source of drinking water, and ownership of dwelling.

During pre-intervention, 228 (65.1%) of the participants never attended school, 132 (37.7%) were pastoralists, 289 (82.6%) of the households had no toilet facility, 237 (67.7%) of the households had thatch/palm leaf/makuti roof, 275 (78.6%) of the households had earth/mud/dung/sand floor, 129 (36.9%) of the households had mud/dung wall, 256 (73.1%) of the households use unimproved water sources for drinking, and 282 (80.6%) of the households were owned by the family.

During post-intervention, 224 (63.8%) of the participants never attended school, 108 (30.8%) were pastoralists, 265 (75.5%) of the households had no toilet facility, 256 (72.9%) of the households had thatch/palm leaf/makuti roof, 287 (81.8%) of the households had earth/mud/dung/sand floor, 149 (42.5%) of the households had mud/dung wall, 298 (90.0%) of the households use unimproved water sources for drinking, and 263 (74.9%) of the households were owned by the family.

**Table 3. Selected socio-demographic characteristics of participants surveyed in Loyamorok Ward, Tiaty East, Baringo County.**

| Characteristic | Number of participants | |
|---|---|---|
| | Pre-intervention (n = 350) | Post-intervention (n = 351) |
| Gender | | |
| Male | 136 (38.9) | 166 (47.3) |
| Female | 214 (61.1) | 185 (52.7) |
| Median age (IQR, Min - Max) | 36 (22, 16-101) | 36 (21, 16 - 93) |
| Marital Status | | |
| Single/Divorced/Widowed | 89 (25.4) | 100 (28.5) |
| Currently married | 261 (74.6) | 251 (71.5) |
| Religion | | |
| Christian | 202 (57.7) | 206 (58.7) |
| Non-practicing | 143 (40.9) | 138 (39.3) |
| Others | 5 (1.4) | 6 (2.0) |

**Table 4. The univariable analysis of selected socio-demographic factors associated with access to mass drug administration for trachoma among participants surveyed in Loyamorok Ward, Tiaty East, Baringo County.**

| Selected socio-demographic factors | N = 701 n (%) | Likelihood of MDA access | | | | Intervention impact [Unadjusted OR (95% CI)], p-value |
|---|---|---|---|---|---|---|
| | | Pre-intervention | | Post-intervention | | |
| | | [Unadjusted OR (95% CI)], p-value | No. of participants | [Unadjusted OR (95% CI)], p-value | No. of participants | |
| Sex | | | | | | |
| Male | 302 (43.1) | 1.27 (0.82-1.95), p=0.283 | 69 | Reference | 114 | Reference |
| Female | 399 (56.9) | Reference | 96 | 1.23 (0.78-1.95), p=0.376 | 135 | 1.56 (0.83-2.93), p=0.168 |
| Age group (years) | | | | | | |
| <20 | 23 (3.3) | Reference | 4 | 2.79 (0.52-14.96), p=0.232 | 9 | 4.50 (0.56-35.84), p=0.155 |
| 20-30 | 182 (26.0) | 2.04 (0.58-7.26), p=0.269 | 47 | 2.44 (1.03-5.79), p=0.043* | 71 | 1.93 (0.73-5.12), p=0.186 |
| 30-40 | 189 (27.0) | 1.91 (0.54-6.81), p=0.317 | 44 | 1.42 (0.63-3.21), p=0.395 | 69 | 1.20 (0.47-3.06), p=0.699 |
| 40-50 | 126 (18.0) | 2.00 (0.54-7.36), p=0.297 | 30 | 1.24 (0.52-2.93), p=0.627 | 44 | Reference |
| 50-60 | 103 (14.7) | 1.52 (0.40-5.69), p=0.537 | 22 | 1.27 (0.52-3.14), p=0.598 | 35 | 1.36 (0.46-3.99), p=0.579 |
| >60 | 78 (11.0) | 1.38 (0.36-5.30), p=0.635 | 18 | Reference | 21 | 1.17 (0.36-3.74), p=0.795 |
| Marital status | | | | | | |
| Single/Divorced/Widowed | 189 (27.0) | Reference | 39 | Reference | 59 | Reference |
| Currently married | 512 (73.0) | 1.20 (0.74-1.94), p=0.467 | 126 | 2.16 (1.32-3.54), p=0.002* | 190 | 1.81 (0.91-3.61), p=0.092 |
| Religion | | | | | | |
| Christian | 408 (58.2) | 4.24 (0.47-38.64), p=0.200 | 104 | 24.15 (2.83-206.15), p=0.004* | 165 | 5.69 (0.26-123.57), p=0.268 |
| Non-practicing | 281 (40.1) | 2.89 (0.32-26.53), p=0.348 | 60 | 9.05 (1.06-77.29), p=0.044* | 83 | 3.13 (0.14-68.39), p=0.468 |
| Others | 12 (1.7) | Reference | 1 | Reference | 1 | Reference |

*Indicates a statistically significant association at 5% level of significance.

Toilet facility coverage was low during both surveys at 61 (17.4%) and 86 (24.5%) during pre-intervention and post-intervention. Particularly, the use of improved toilet was 39 (63.9%) during pre-intervention and 20 (23.3%) during post-intervention.

Assessment of the household structures, during both surveys, qualified the majority of the households to temporary structure category. Over three-quarters, 638 (91.0%), of the households reportedly spent up to more than 15 mins to fetch water from the nearest water source, with the availability of water at that source being mostly infrequent, 282 (40.2%).

Table 7 assessed the association between selected socio-economic factors and access to trachoma drugs using univariable analysis. Assessment of the intervention impact (changes in odds ratio pre- and post-intervention), indicated that occupation of the participant was the main factor significantly associated with the increased MDA access

**Table 5. The multivariable analysis of selected socio-demographic factors associated with access to mass drug administration for trachoma among participants surveyed in Loyamorok Ward, Tiaty East, Baringo County.**

| Selected socio-demographic factors | N = 701 n (%) | Likelihood of MDA access | | | | Intervention impact [Adjusted OR (95% CI)], p-value |
|---|---|---|---|---|---|---|
| | | Pre-intervention | | Post-intervention | | |
| | | [Adjusted OR (95% CI)], p-value | No. of participants | [Adjusted OR (95% CI)], p-value | No. of participants | |
| Marital status | | | | | | |
| Single/Divorced/Widowed | 189 (27.0) | Reference | 39 | Reference | 59 | Reference |
| Currently married | 512 (73.0) | 1.17 (0.72-1.90), p = 0.539 | 126 | 2.01 (1.21-3.36), p = 0.007* | 190 | 1.73 (0.85-3.50), p = 0.129 |
| Religion | | | | | | |
| Christian | 408 (58.2) | 4.15 (0.45-37.84), p = 0.207 | 104 | 24.81 (2.87-214.27), p = 0.004* | 165 | 5.98 (0.27-131.19), p = 0.256 |
| Non-practicing | 281 (40.1) | 2.84 (0.31-26.09), p = 0.356 | 60 | 9.93 (1.15-85.87), p = 0.037* | 83 | 3.50 (0.16-77.17), p = 0.428 |
| Others | 12 (1.7) | Reference | 1 | Reference | 1 | Reference |

*Indicates a statistically significant association at 5% level of significance.

for trachoma, housewife participants (OR = 9.03, 95% CI: 1.03-78.87, p = 0.046) and pastoralists (OR = 9.23, 95% CI: 1.23-69.00, p = 0.030) showed significantly increased access to MDA compared to salaried workers. Additionally, participants from houses with floors made of wood/palm/bamboo also showed significantly increased access to MDA (OR = 17.42, 95% CI: 2.08-145.84, p = 0.008) compared to participants from houses with floors made of cement/tiles/carpet.

Table 8 assessed the association between selected socio-economic factors and access to trachoma drugs using multivariable analysis. Assessment of the intervention impact (changes in odds ratio pre- and post-intervention) indicated that occupation of the participant, type of house floor and wall materials were the main factors significantly associated with the increased MDA access for trachoma. Housewife participants (aOR = 12.87, 95% CI: 1.09-151.52, p = 0.042) and pastoralists (aOR = 11.45, 95% CI: 1.15-114.14, p = 0.038) showed significantly increased access to MDA compared to salaried workers. Participants from houses with floors made of earth/mud/dung (aOR = 22.69, 95% CI: 2.77-186.18, p = 0.004) and wood/palm/bamboo (aOR = 165.90, 95% CI: 8.57-3211.19, p = 0.001) showed significantly increased access to MDA compared to floors made of cement/tiles/carpet. Additionally, participants from houses with walls made of cement/block/stone/brick (aOR = 9.54, 95% CI: 1.65-55.33, p = 0.012) showed significantly increased access to MDA compared to walls made of wood/palm/bamboo.

## Participants' knowledge about trachoma

During the pre-intervention survey, just slightly more than a half 192 (54.9%) of the respondents had knowledge about trachoma, with the other proportion of the respondents 158 (45.1%) reporting that they had never heard about it. The respondents who reported knowing trachoma, reportedly on average know at least 3 people (SD = 4.6 people) who are "infected". However, only less than a half 164 (46.9%) of the respondents correctly stated that it is caused by flies. Respondents who had no idea of what causes it were 106 (30.3%), while 36 (10.3%) cited other causes like dust, dirty water, inheritance, and old age, among others (Fig 1). Over half 196 (56.0%) of the respondents acknowledged that they are at risk of getting trachoma.

**Table 6. Selected socio-economic characteristics of participants surveyed in Loyamorok Ward, Tiaty East, Baringo County.**

| Selected socio-economic factors | Number of participants | |
|---|---|---|
| | Pre-intervention n (%), n = 350 | Post-intervention n (%), n = 351 |
| Education level | | |
| Never attended school | 228 (65.1) | 224 (63.8) |
| Primary | 69 (19.7) | 75 (21.4) |
| Secondary | 36 (10.3) | 41 (11.6) |
| Post-secondary | 17 (4.9) | 11 (3.1) |
| Occupation | | |
| Business | 98 (28.0) | 49 (14.0) |
| Housewife | 30 (8.6) | 47 (13.4) |
| Salaried worker | 11 (3.1) | 13 (3.7) |
| | 132 (37.7) | 108 (30.8) |
| Farmer | 42 (12.0) | 84 (23.9) |
| Casual laborer | 25 (7.1) | 43 (12.3) |
| Others | 12 (3.4) | 7 (2.0) |
| Presence of toilet facility | | |
| Yes | 61 (17.4) | 86 (24.5) |
| No | 289 (82.6) | 265 (75.5) |
| Toilet type | | |
| Unimproved | 22 (36.1) | 66 (76.7) |
| Improved | 39 (63.9) | 20 (23.3) |
| Type of roofing material | | |
| Brick/Concrete/Tiles | 4 (1.1) | 9 (2.6) |
| Corrugated iron sheet | 96 (27.4) | 78 (22.2) |
| Thatch/Palm leaf/Makuti | 237 (67.7) | 256 (72.9) |
| Others | 13 (3.7) | 8 (2.3) |
| Type of flooring material | | |
| Earth/Mud/Dung/Sand | 275 (78.6) | 287 (81.8) |
| Wood/Palm/Bamboo | 13 (3.7) | 15 (4.3) |
| Cement/Tiles/Carpet | 62 (17.7) | 45 (12.8) |
| Others | 0 | 4 (1.1) |
| Type of wall material | | |
| Cement/Block/Stone/Brick | 107 (30.6) | 62 (17.7) |
| Corrugated iron sheet | 6 (1.7) | 9 (2.6) |
| Mud/Dung | 129 (36.9) | 149 (42.5) |
| Wood/Palm/Bamboo | 108 (30.9) | 131 (37.3) |
| Source of drinking water | | |
| Unimproved | 256 (73.1) | 298 (90.0) |
| Improved | 94 (26.9) | 33 (10.0) |
| Ownership of dwelling | | |
| Owned by family | 282 (80.6) | 263 (74.9) |
| Rented | 4 (1.1) | 5 (1.4) |
| No rent, with owners consent | 44 (12.6) | 74 (21.1) |
| No rent, squatting | 19 (5.4) | 8 (2.3) |
| Others | 1 (0.3) | 1 (0.3) |

**Table 7. Univariable analysis of selected socio-economic factors associated with access to mass drug administration for trachoma among participants surveyed in Loyamorok Ward, Tiaty East, Baringo County.**

| Selected socio-economic factors | N = 701 n (%) | Likelihood of MDA access [Unadjusted OR (95% CI)], p-value | | Intervention impact [Unadjusted OR (95% CI)], p-value |
|---|---|---|---|---|
| | | **Pre-intervention** | **Post-intervention** | |
| Education level | | | | |
| Never attended school | 452 (64.5) | Reference | Reference | 1.23 (0.23-6.61), p=0.805 |
| Primary complete | 144 (20.5) | 1.60 (0.93-2.75), p=0.090 | 1.31 (0.73-2.37), p=0.369 | 1.01 (0.17-6.01), p=0.987 |
| Secondary complete | 77 (11.0) | 1.46 (0.72-2.95), p=0.296 | 1.21 (0.58-2.56), p=0.611 | 1.03 (0.16-6.81), p=0.976 |
| Post-secondary | 28 (4.0) | 1.47 (0.54-3.94), p=0.448 | 1.19 (0.31-4.61), p=0.804 | Reference |
| Occupation | | | | |
| Business | 6 (0.9) | 1.73 (0.74-4.01), p=0.203 | Reference | 1.63 (0.21-12.68), p=0.641 |
| Casual laborer | 68 (9.7) | 1.36 (0.46-4.01), p=0.581 | 2.83 (1.20-6.70), p=0.018* | 5.87 (0.66-52.32), p=0.113 |
| Farmer | 126 (18.0) | 2.30 (0.88-6.03), p=0.089 | 5.22 (2.38-11.42), p<0.001* | 6.38 (0.78-52.00), p=0.084 |
| Housewife | 77 (11.0) | Reference | 3.21 (1.37-7.52), p=0.007* | 9.03 (1.03-78.87), p=0.046* |
| Pastoralist | 240 (34.2) | 1.12 (0.49-2.55), p=0.782 | 3.68 (1.81-7.50), p<0.001* | 9.23 (1.23-69.00), p=0.030* |
| Salaried worker | 24 (3.4) | 7.77 (1.42-42.66), p=0.018* | 2.76 (0.75-10.19), p=0.127 | Reference |
| Others | 19 (2.7) | 5.18 (1.15-23.29), p=0.032* | 3.07 (0.54-17.37), p=0.205 | 1.67 (0.10-28.86), p=0.726 |
| Toilet facility | | | | |
| No facility | 554 (79.0) | Reference | Reference | 2.54 (0.76-8.50), p=0.132 |
| Unimproved | 88 (12.6) | 4.52 (1.62-12.60), p=0.004* | 4.32 (0.98-19.06), p=0.053 | Reference |
| Improved | 59 (8.4) | 2.13 (1.07-4.23), p=0.031* | 1.78 (0.94-3.40), p=0.078 | 5.15 (0.72-37.03), p=0.104 |
| Roof material | | | | |
| Thatch/Palm leaf/Makuti | 493 (70.3) | 2.61 (0.70-9.71), p=0.153 | Reference | 3.75 (0.26-53.77), p=0.330 |
| Iron sheet | 174 (24.8) | 4.47 (1.16-17.28), p=0.030* | 1.19 (0.67-2.10), p=0.549 | 2.60 (0.17-39.25), p=0.490 |
| Brick/Concrete/Tiles | 14 (2.0) | 10.00 (0.74-135.33), p=0.083 | 1.02 (0.26-4.06), p=0.975 | Reference |
| Others | 20 (2.9) | Reference | Insufficient observation | Insufficient observation |
| Floor material | | | | |
| Earth/Mud/Dung/Sand | 562 (80.2) | 4.58 (1.00-21.07), p=0.049* | 1.23 (0.63-2.40), p=0.547 | 2.33 (0.97-5.60), p=0.058 |
| Wood/Palm/Bamboo | 28 (4.0) | Reference | 2.00 (0.49-8.18), p=0.335 | 17.42 (2.08-145.84), p=0.008* |
| Cement/Tiles/Carpet | 107 (15.3) | 8.71 (1.77-42.74), p=0.008* | Reference | Reference |
| Others | 4 (0.6) | Insufficient observation | 1.50 (0.14-15.67), p=0.735 | Insufficient observation |
| Wall material | | | | |
| Cement/Block/Stone/Brick | 169 (24.1) | 1.72 (1.00-2.96), p=0.049* | 2.33 (1.15-4.71), p=0.019* | 1.35 (0.56-3.29), p=0.507 |
| Iron sheet | 15 (2.1) | 2.91 (0.51-16.58), p=0.229 | 4.94 (0.60-40.67), p=0.138 | 1.70 (0.11-26.13), p=0.704 |
| Mud/Dung | 278 (39.7) | 1.23 (0.73-2.06), p=0.440 | 1.80 (1.08-3.00), p=0.023* | 1.47 (0.71-3.04), p=0.298 |
| Wood/Palm/Bamboo | 239 (34.1) | Reference | Reference | Reference |
| Cooking fuel | | | | |
| Electricity/Gas | 7 (1.0) | 2.26 (0.20-25.13), p=0.508 | Reference | Reference |
| Firewood/Charcoal/Kerosene | 694 (99.0) | Reference | 2.47 (0.34-17.78), p=0.369 | 5.58 (0.25-125.63), p=0.280 |
| Drinking water source | | | | |
| Unimproved | 127 (18.7) | 1.21 (0.75-1.95), p=0.424 | 1.87 (0.89-3.93), p=0.101 | 1.54 (0.63-3.72), p=0.341 |
| Improved | 554 (81.4) | Reference | Reference | Reference |

*Indicates a statistically significant association at 5% level of significance.

**Table 8. Multivariable analysis of selected socio-economic factors associated with access to mass drug administration for trachoma among participants surveyed in Loyamorok Ward, Tiaty East, Baringo County.**

| Selected socio-economic factors | N=701 n (%) | Likelihood of MDA access [Adjusted OR (95% CI)], p-value | | Intervention impact [Adjusted OR (95% CI)], p-value |
|---|---|---|---|---|
| | | Pre-intervention | Post-intervention | |
| Occupation | | | | |
| Business | 6 (0.9) | 1.34 (0.56-3.24), p=0.510 | Reference | 2.10 (0.20-21.65), p=0.534 |
| Casual laborer | 68 (9.7) | 1.05 (0.34-3.27), p=0.933 | 3.91 (1.39-10.95), p=0.010* | 10.49 (0.86-128.55), p=0.066 |
| Farmer | 126 (18.0) | 1.82 (0.67-4.95), p=0.241 | 5.27 (2.07-13.41), p<0.001* | 8.16 (0.75-88.51), p=0.084 |
| Housewife | 77 (11.0) | Reference | 4.57 (1.74-11.96), p=0.002* | 12.87 (1.09-151.52), p=0.042* |
| Pastoralist | 240 (34.2) | 1.17 (0.49-2.77), p=0.722 | 4.75 (2.10-10.75), p<0.001* | 11.45 (1.15-114.14), p=0.038* |
| Salaried worker | 24 (3.4) | 6.09 (1.04-35.84), p=0.046* | 2.16 (0.41-11.46), p=0.365 | Reference |
| Others | 19 (2.7) | 4.63 (0.99-21.62), p=0.052 | Insufficient observations | Insufficient observations |
| Toilet facility | | | | |
| No facility | 554 (79.0) | Reference | Reference | Reference |
| Unimproved | 88 (12.6) | 4.11 (1.35-12.45), p=0.013* | Insufficient observations | 0.89 (0.20-3.94), p=0.874 |
| Improved | 59 (8.4) | 1.83 (0.74-4.54), p=0.193 | 3.64 (1.34-9.87), p=0.011* | Insufficient observations |
| Floor material | | | | |
| Earth/Mud/Dung/Sand | 562 (80.2) | 3.81 (0.80-18.14), p=0.093 | 16.16 (2.37-110.20), p=0.004* | 22.69 (2.77-186.18), p=0.004* |
| Wood/Palm/Bamboo | 28 (4.0) | Reference | 30.98 (2.85-337.13), p=0.005* | 165.90 (8.57-3211.19), p=0.001* |
| Cement/Tiles/Carpet | 107 (15.3) | 5.35 (0.93-30.95), p=0.061 | Reference | Reference |
| Others | 4 (0.6) | Insufficient observation | Insufficient observation | Insufficient observation |
| Wall material: | | | | |
| Cement/Block/Stone/Brick | 169 (24.1) | 0.85 (0.41-1.79), p=0.674 | 8.14 (1.65-40.08), p=0.010* | 9.54 (1.65-55.33), p=0.012* |
| Iron sheet | 15 (2.1) | 1.53 (0.24-9.67), p=0.650 | Insufficient observations | Insufficient observations |
| Mud/Dung | 278 (39.7) | 1.16 (0.66-2.03), p=0.609 | 1.56 (0.86-2.81), p=0.144 | 1.34 (0.59-3.04), p=0.479 |
| Wood/Palm/Bamboo | 239 (34.1) | Reference | Reference | Reference |
| Cooking fuel | | | | |
| Electricity/Gas | 7 (1.0) | Reference | Reference | Reference |
| Firewood/Charcoal/Kerosene | 694 (99.0) | 1.15 (0.09-15.64), p=0.914 | 2.47 (0.34-17.78), p=0.369 | 1.15 (0.09-15.64), p=0.914 |
| Drinking water source | | | | |
| Unimproved | 127 (18.7) | 1.45 (0.86-2.45), p=0.168 | 3.08 (1.20-7.93), p=0.020* | 2.13 (0.72-6.27), p=0.172 |
| Improved | 554 (81.4) | Reference | Reference | Reference |

*Indicates a statistically significant association at 5% level of significance.

During the post-intervention survey, a slightly reduced number of participants 146 (41.6%) reported knowledge of anyone with trachoma. They reported knowing at least 2 people (SD = 3.6 people) who are infected. Majority 230 (65.5%) correctly cited that trachoma is caused by flies (Fig 1). There was an improved number of participants reporting the correct cause of trachoma. More than half of the participants 197 (56.1%) considered themselves at risk of contracting trachoma.

## Participants' knowledge about mass drug administration for trachoma

During the pre-intervention phase, the results showed that over half 190 (54.3%) of the respondents had never heard about MDA for elimination of trachoma in their community. While the minority (45.7%) who had heard about MDA, learnt

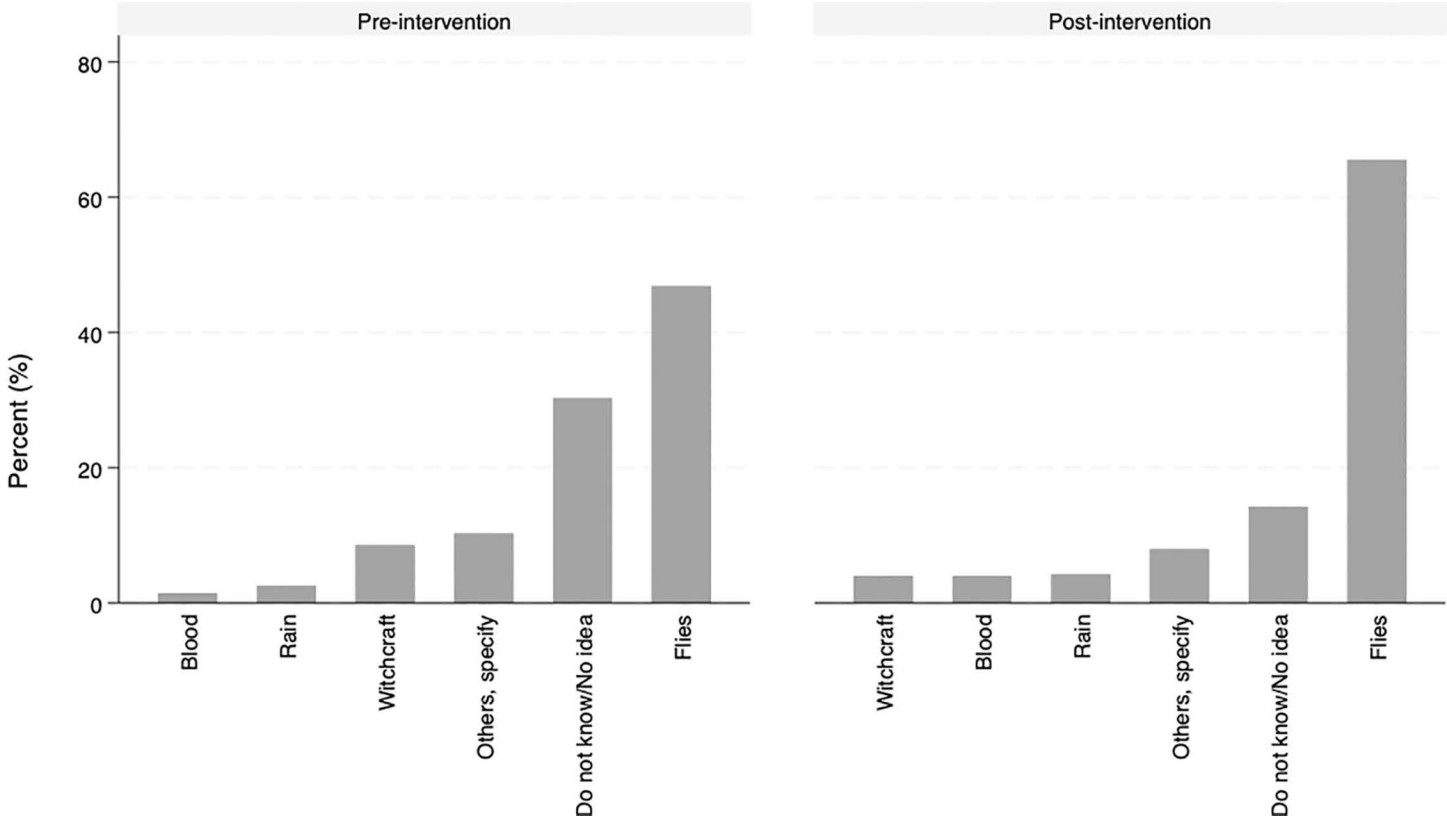

**Fig 1. Reported causes of trachoma among participants surveyed in Loyamorok Ward, Tiaty East, Baringo County.**

about it through chief's meetings (29.6%), community health workers (25.9%), health facility (11.1%), community health volunteers (9.3%), and radio and other media channels (3.7%), among other platforms (Fig 2). During previous MDAs, participants reported that they were informed about trachoma elimination (26.5%), dosage (15.5%), importance of taking drugs (14.8%), what the drug treats (9.0%), treatment eligibility (7.7%), drug distribution strategies (5.2%), available trachoma drugs (3.9%), MDA campaigns (3.2%), and prevention and causes of trachoma (1.3%), among other information (2.6%). However, 9.0% of the participants reported that they were not given any additional information during MDA implementation.

During the post-intervention study, a non-significantly increased number of participants 187 (53.3%) reportedly were aware of MDA (Table 9), while 164 (46.7%) had not heard about it. For those aware of MDA (n = 187), marketplaces 17 (9.1%), town criers 11 (5.9%), community health workers (CHWs) 10 (5.3%), chief's meetings 10 (5.3%) and health facilities 6 (3.2%) were significant sources of the information about the MDA (Fig 2). Whereas posters 1 (0.5%), radio and other media channels 4 (2.1%), and village elders 1 (0.5%), contributed insignificantly to the MDA awareness. Participants who heard about MDA (n = 187) were asked about the information they acquired and treatment eligibility 25 (13.4%), drugs administration 27 (14.4%), and trachoma elimination 15 (8.0%), drug dosage 12 (6.4%) and drug delivery 12 (6.4%) were prominent while only 5 (2.7%) heard about the treatment regimen, 3 (1.6%) heard about causes of trachoma and 2 (1.1%) heard about the potential side effects of the drug.

## Sources of information about mass drug administration

When asked how frequent they get information about MDA during the pre-intervention phase, most participants reported getting the information once a year (82.9%), twice a year (7.0%), multiple times within a year (3.2%), while another 3.2%

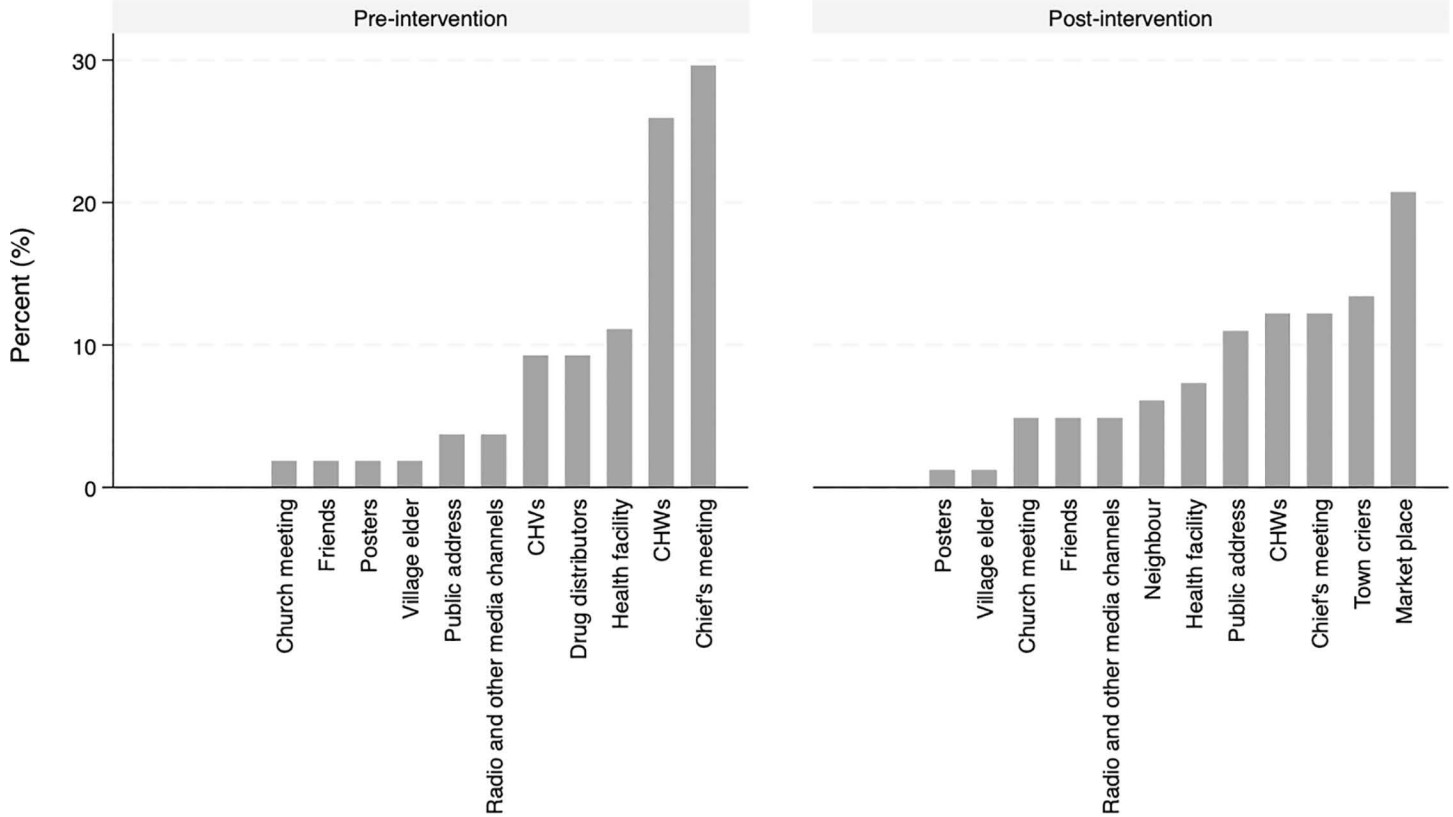

**Fig 2. Reported channels of communication of mass drug administration for elimination of trachoma in Loyamorok Ward, Tiaty East, Baringo County.**

**Table 9. Assessment of participants' knowledge about mass drug administration for trachoma in Loyamorok Ward, Tiaty East, Baringo County.**

| Participants' knowledge outcomes | Pre-interventions (n = 350) | Post-interventions (n = 351) | Difference between pre- and post-interventions | Overall (n = 701) |
|---|---|---|---|---|
| Proportion of participants who have heard about MDA | 160 (45.7) | 187 (53.3) | Diff = -0.076, z = -1.41, p = 0.158 | 347 (49.5) |
| Most common channel through which they heard about MDA | Chief's meeting | Market place | – | – |
| Proportion of participants reached by this channel | 16 (29.6) | 17 (20.7) | Diff = 0.089, z = 0.59, p = 0.555 | 24 (17.7) |

got it infrequently (after more than one year) and 3.8% can't remember. When the participants were asked about their opinion regarding the source of the information about MDA, majority (59.5%) rated the source as good and effective since it reaches many people and preferred continuing with the same source. However, other participants preferred change of the source (2.0%), use chiefs/ village elders/ *wazee wa nyumba kumi* (8.5%), use public barazas/ meetings (2.6%), and consider more planning (0.7%), among other information.

During the post-intervention phase, participants reported varying frequencies of acquiring information about MDA with majority being those who reported once a year 60 (32.1%), followed by weekly 28 (15.0%), twice in a year 24 (12.8%), daily 20 (10.7%), thrice a year 17 (9.1%) and 10 (5.3%) reported to have acquired the information frequently. Regarding the participant's opinion on the source of information during the last MDA (n = 187), majority of the participants reported

that it was effective and well done 108 (57.8%) while 22 (11.8%) felt there was room for change whereas 10 (5.3%) felt that there ought to be an extension of time for awareness and 18 (9.6%) reported to have no opinion on the source of information.

During the pre-intervention study, nearly half of the participants were reportedly told that the trachoma drugs were safe and had no possible side effects (40.1%), however, others reportedly received varied information about the possible side effects of the drugs such as vomiting and nausea (15.8%), constipation (5.3%), stomachache (3.3%), among other information (13.8%). While 10.5% reported that they didn't receive any information or can't remember the possible side effects. However, it was surprising that some participants (4.0%) were told that taking drugs could lead to blindness or eye problems.

In the post-intervention phase, 31 (16.6%) respondents were reportedly told that the drugs had no side effects whereas majority of the participants 77 (41.2%) were reportedly told that vomiting and diarrhoea were the main side effects followed by stomachache 9 (4.8%), headache 8 (4.3%), heartburn 8 (4.3%), dizziness 4 (2.1%) and skin rush 4 (2.1%).

## Mass drug administration awareness creation

Participants were asked about how they want treatment awareness to be created during the next round of MDA in the pre-intervention stage, they suggested varied awareness creation strategies that included public meetings, barazas and peacekeeping meetings (32.7%), use of chiefs, village elders and *wazee wa nyumba kumi* (15.4%), use of house to house approach (10.3%), use of radio and other mass media channels (8.3%), use of CHWs and community health volunteers (CHVs) (6.4%), use of health facilities and schools (3.2%), and use of public places like churches, water points and market places (3.2%). However, 5.8% of the respondents preferred retaining the previous strategy, while 5.1% had no idea of how the MDA awareness should be created. Further, participants suggested varied timelines within which MDA awareness should be created prior to the actual MDA delivery; within 1 week before MDA (39.6%), within 2 weeks before MDA (31.2%), within 3 weeks or more before MDA (29.2%).

During the post-intervention stage, participants expressed preferences for the next MDA awareness creation method whereby 38 (20.3%) preferred maintaining the previous method, while 35 (18.7%) preferred Chief barazas and village elders. Mass media and posters and public places were each reported by 26 (13.9%) while 15 (8.0%) preferred house to house and village to village method. Regarding the duration for awareness creation, majority of the participants preferred two weeks 52 (27.8%) followed by one week 41 (21.9%), one month 24 (12.8%), three weeks 14 (7.5%) and less than one week and more than one month were each reported by 8 (4.3%).

## Reported uptake of trachoma drugs during previous mass drug administration

During the pre-intervention study, only close to two thirds 228 (65.1%) of the participants reported that they had ever taken trachoma drugs. Out of the 228 participants, 165 (72.4%) reported taking the drugs during the last MDA. The surveyed participants had on average taken trachoma drugs for two consecutive times (MDA years) (range: 1–7 times). However, those who had never taken drugs majorly argued that they were away attending to domestic chores and grazing animals (28.7%), they feared the insecurity situation (23.0%), away on family visits or attending to their businesses (14.8%), they were in the move searching for pasture for the animals (11.5%), harsh geographical terrain (9.8%), among other reasons like lack of prior information, fear of side effects, and not reached with the drug.

In the post-intervention phase, a significantly increased number of participants 268 (76.4%) reported having ever taken drugs during MDA (Diff = -0.113, z = -2.770, p = 0.006), with a significant increased number 249 (92.9%) of participants reporting having taken drugs during the last MDA (Diff = -0.205, z = -5.68, p < 0.001) (Table 10). Out of the 268 participants who took MDA drugs, 57 (21.3%) took the drugs once while a substantial proportion took the drugs more than once, with 93 (34.7%) taking twice, 65 (24.3%) taking thrice, 9 (3.4%) taking four times, 25 (9.3%) taking five times and 19 (7.1%)

taking eight times. For those who did not take drugs during MDA (n = 83), the main reported reason for non-participation was absence from household 60 (72.3%), followed by geographical terrain 11 (13.3%) and insecurity 7 (8.4%).

### Participants' perceptions on treatment for trachoma

During the pre-intervention study, majority (84.9%) of the participants considered the treatment as necessary, and had no problem with swallowing pills (88.6%), size of the pills (94.6%), number of the pills (94.6%), or taste of the pills (91.1%), and indicated that they would take the drugs again (79.4%). However, 9.7% of the respondents affirmed that they would not take the drugs again mainly due to the fear of the reactions or side effects, they considered the drugs as not necessary to them, insecurity in the area, or will be away looking after animals.

In the post-intervention study, a significant majority of participants 277 (78.9%) acknowledged the necessity of treatment while a subset of participants faced specific challenges during drug intake, with 51 (14.5%) experiencing trouble swallowing pills, 35 (10.0%) having issues with the size of pills, 28 (8.0%) having problems with the number of pills given, and 22 (6.3%) expressing dissatisfaction with the taste of the pills. The majority of participants 276 (78.6%) expressed willingness to take the drugs next time they are administered while 25 (7.1%) were not willing to take the drugs during the next administration, and 49 (14.0%) were uncertain. Out of the 25 participants who were unwilling to take the drugs when next administered, 6 (24.0%) reported dislike for modern medicine, 5 (20.0%) reported being absent during distribution, 5 (20.0%) reported fear of adverse reactions, 4 (16.0%) reported that the drugs were unnecessary for them and 3 (12.0%) reported insecurity in the area.

### Suggested strategies on how mass drug administration should be conducted

During pre-intervention study, only three-fifths, 215 (61.4%) of the participants preferred the drugs to be distributed the same way during subsequent MDAs. However, about a fifth 63 (18.0%) preferred a change of strategy in the distribution of the drugs. They cited that CHWs who were distributing drugs did not clearly explain the need to take the drugs and the associated side effects (27.0%), they had to wait for long hours before being given the drugs (23.8%), they had poor interaction experience with the CHWs (22.2%), or CHWs did not have enough drugs to give (15.9%).

During post-intervention, more than half of the participants 272 (77.5%) preferred drugs to be administered in a similar way next time whereas 21 (6.0%) did not want drugs administered in the same manner with some of the reasons being cited including poor interaction with CHWs 7 (33.3%), long waiting hours for CHWs 7 (33.3%), insufficient drugs 2 (9.5%) and unclear explanations by CHWs 1 (4.8%).

**Table 10. Uptake of trachoma drugs and participants' perceptions on treatment in Loyamorok Ward, Tiaty East, Baringo County.**

| Drug uptake outcomes | Pre-interventions (n = 350) | Post-interventions (n = 351) | Difference between pre- and post-interventions | Overall (n = 701) |
|---|---|---|---|---|
| Proportion ever taken trachoma drugs | 228 (65.1) | 268 (76.4) | Diff = -0.113, z = -2.770, p = 0.006* | 496 (70.8) |
| Proportion who took trachoma drugs during last MDA | 165 (72.4) | 249 (92.9) | Diff = -0.205, z = -5.68, p < 0.001* | 414 (83.5) |
| Number of times taken trachoma drugs (mean; range)* | 2.0; 1-7 | 2.8; 1-8 | Diff = -0.800, t = -6.229, p < 0.001* | 2.4; 1-8 |
| Proportion who considers treatment as necessary | 297 (84.9) | 277 (78.9) | Diff = 0.06, z = 1.87, p = 0.062 | 574 (81.9) |
| Proportion who expressed problems swallowing drugs | 40 (11.4) | 51 (14.9) | Diff = 0.012, z-0.18, p = 0.854 | 91 (13.1) |
| Proportion who expressed problem with size of drugs | 19 (5.4) | 35 (10.2) | Diff = -0.048, z = -0.60, p = 0.546 | 54 (7.8) |
| Proportion who expressed problem with number of drugs | 19 (5.4) | 28 (8.2) | Diff = -0.028, z = -0.37, p = 0.713 | 47 (6.8) |
| Proportion who expressed problem with taste of drugs | 31 (8.9) | 22 (6.5) | Diff = 0.024, z = 0.32, p = 0.750 | 53 (7.7) |
| Proportion who affirmed intent to take trachoma drugs in future MDA | 278 (79.4) | 276 (78.9) | Diff = 0.005, z = 0.14, p = 0.885 | 554 (79.1) |

*The difference in the mean number of times an individual took trachoma drugs was assessed using two-sample t-test at 95% confidence interval.

When asked what distribution method should be used in subsequent MDAs during the pre-intervention study, nearly half (45.1%) preferred door to door distribution method, over a quarter (27.2%) preferred that the drugs to be administered at a central point in a village (e.g., in health facilities, schools, markets, churches or water points), 14.2% of the respondents preferred drugs to be administered by chiefs and village elders in every village, the remaining respondents preferred other methods like use of CHWs and CHVs, administration according to specific groups, mobile distribution, among other methods. On the choice of distributors, majority of the participants (61,4%) preferred the continued use of CHWs and CHVs but accompanied by village elders, 13.5% preferred youths from their village to distribute the drugs, while 12.7% preferred the public health officials. On the distribution duration, most participants preferred a distribution period of between one week and one month. On the time of distribution, participants preferred distribution time of between 8am to 5pm, and during rainy season when everybody is around.

In the post-intervention phase, majority of the participants preferred house-to-house drug distribution method 205 (58.4%), followed by distribution through marketplaces 24 (6.8%) and the preferred drug distributors were CHWs 112 (31.9%) and CHVs 106 (30.2%) followed by health officers 65 (18.5%) with the most preferred drug distribution duration being two weeks 93 (26.5%), followed by one week 81 (23.1%) and one month 68 (19.4%). The most preferred drug distribution timing was during the rainy season 169 (48.1%) followed by morning hours 64 (18.2%) because those are the times when people are present in their houses.

## Discussion

Community-participatory approaches, including effective stakeholder coordination, community awareness, and strategic planning, have been proposed as strategies to improve access to MDA in challenging settings like pastoral conflict areas [15]. However, the effectiveness of these approaches in enhancing MDA uptake remains uncertain. This study aimed to evaluate the impact of community-participatory approaches on improving access to MDA for trachoma elimination in a pastoral conflict area of Baringo County, Kenya. Our findings revealed that occupation, effective communication, and enhanced capacity building of MDA campaign teams are significant factors influencing access to MDA.

The study identified occupation and type of house flooring as significant socioeconomic factors influencing access to MDA for trachoma. Notably, individuals from lower socioeconomic backgrounds, such as housewives, pastoralists, and those from houses made of wood/palm/bamboo floors, demonstrated higher rates of MDA access, similar results have also been obtained in other studies [16]. This trend can be attributed to the hypothesis that individuals from lower socioeconomic classes have greater reliance on free health services. In contrast, individuals from higher socioeconomic classes, particularly salaried workers, who may have the financial means to seek private healthcare, exhibited lower levels of MDA participation. In this study, MDA access refers to the number of individuals who took the drugs (offered with the antibiotics for trachoma and swallowed the drugs) during an MDA exercise.

The findings of this study further indicated a significant knowledge gap among community members regarding trachoma. Individuals with limited knowledge often place a lower priority on disease prevention [17]. The strategies implemented and evaluated in this study were effective in enhancing knowledge about trachoma. The results suggest the need for ongoing health education and repeated awareness campaigns to promote behaviour change and emphasize the cause and prevention of the disease. These initiatives should not be restricted to MDA campaign periods but should be conducted more regularly. Providing information about disease transmission factors can encourage greater community participation in MDA, as observed in previous research [15].

The study findings revealed that CHWs involved in drug distribution during the MDA exercise had poor interaction with the participants. Additionally, participants cited that CHWs were not clearly explaining the need to take the drugs. This underscores the necessity of investing in CHWs training to enhance their effectiveness. Research shows that inadequately trained volunteers may feel overwhelmed when unable to confidently address community members' questions [18]. It is also crucial to motivate CHWs, given their multifaceted roles encompassing health education, MDA awareness

promotion, drug distribution, and record-keeping. The study also highlights the importance of allocating sufficient time for community members to gain knowledge and prepare for treatment, a key factor in campaign success that aligns with findings from a study conducted in Luangwa District, Ghana [19].

## Study strengths and limitations

The main limitation of this study is that it did not have a control group, which limits our assessment of the direct cause-and-effect relationship between community participation and better access to trachoma drugs. While the study shows that community involvement is helpful, its exact impact without a comparison group cannot be ascertained. To better understand this, future studies should include a control group to compare results.

Another limitation to this study is that, even though we targeted household heads aged 18 years and above, MDA is usually given to all individuals aged 6 months and above. It was assumed that the views of the individuals aged below 18 years, who were still subject to MDA, were represented by their household heads. This study also relied on self-reported data collected through household surveys, which are susceptible to various biases. One is social desirability bias, where respondents may provide answers they believe are more socially acceptable or aligned with perceived expectations, particularly regarding their participation in health initiatives and their acceptance of MDA. We attempted to mitigate this bias by ensuring the anonymity and confidentiality of responses and training interviewers to maintain neutrality. It is also worth noting that these findings should be interpreted with caution, particularly since some associations present with wide confidence intervals, which suggests low precision of the estimates.

A key strength of this study is that it incorporated community involvement and participation in the co-creation of strategies for trachoma elimination using MDA. This highlights the importance of involving communities in public health programs to give effective intervention strategies.

## Conclusions

This study assessed the impact of community participation strategies on MDA access for trachoma elimination in Baringo County, Kenya. Our findings indicated that socioeconomic factors, particularly occupation, significantly influenced MDA access. Individuals from lower socioeconomic backgrounds were more likely to participate in MDA, potentially due to their reliance on public health services. The study also highlighted the crucial role of health education in increasing community awareness about trachoma and MDA access. By providing accurate information about disease transmission and prevention, community members were more likely to participate in MDA. Chiefs' meetings, marketplaces, and CHWs emerged as some of the key sources for disseminating health information and facilitating MDA uptake. However, challenges such as inadequate training and motivation for drug distributors remain. To optimize the effectiveness of community-based interventions, it is imperative to invest in capacity building and provide drug distributors with the necessary resources and support. Additionally, ongoing health education campaigns are essential to sustain community engagement and maintain high levels of MDA coverage. By addressing these factors, we can significantly improve the impact of trachoma control programs and ultimately achieve the goal of elimination. While the observed improvements in MDA access are associated with the implementation of our community participation strategies, we cannot definitively attribute this increase solely to these interventions. Other factors or secular trends may have contributed to the observed changes. Future research employing more rigorous study designs, such as cluster-randomized trials, is needed to establish a causal relationship and more definitively assess the impact of community participation strategies on MDA access.

## Supporting information

**S1 Data Tool. Data collection instrument used during pre- and post-intervention to collect quantitative data.**
(DOCX)

## Acknowledgments

We acknowledge the support received from the National Trachoma Coordination team, Baringo County Health Management Team (CHMT), Tiaty East Sub-County Health Management Team (SCHMT), and all stakeholders involved for their technical input during the implementation of this study. We are also grateful to the people of Loyamorok Ward for their participation and cooperation during the data collection activities. This study has been published with the permission of the Director General, Kenya Medical Research Institute.

## Author contributions

**Conceptualization:** Collins Okoyo, Bridget W. Kimani, Titus Waititu, Wyckliff P. Omondi, Doris W. Njomo.

**Data curation:** Collins Okoyo, Omar Kopi, Paul M. Gichuki, Bridget W. Kimani, Tabitha Kanyui, Titus Waititu, Wyckliff P. Omondi, Doris W. Njomo.

**Formal analysis:** Collins Okoyo, Omar Kopi.

**Funding acquisition:** Doris W. Njomo.

**Investigation:** Collins Okoyo, Omar Kopi, Paul M. Gichuki, Bridget W. Kimani, Tabitha Kanyui, Titus Waititu, Wyckliff P. Omondi, Doris W. Njomo.

**Methodology:** Collins Okoyo, Paul M. Gichuki, Bridget W. Kimani, Tabitha Kanyui, Titus Waititu, Wyckliff P. Omondi, Doris W. Njomo.

**Project administration:** Collins Okoyo, Omar Kopi, Paul M. Gichuki, Bridget W. Kimani, Tabitha Kanyui, Titus Waititu, Wyckliff P. Omondi, Doris W. Njomo.

**Resources:** Collins Okoyo, Bridget W. Kimani, Doris W. Njomo.

**Software:** Collins Okoyo, Omar Kopi, Doris W. Njomo.

**Supervision:** Collins Okoyo, Omar Kopi, Paul M. Gichuki, Bridget W. Kimani, Tabitha Kanyui, Titus Waititu, Wyckliff P. Omondi, Doris W. Njomo.

**Validation:** Collins Okoyo, Omar Kopi, Paul M. Gichuki, Bridget W. Kimani, Tabitha Kanyui, Titus Waititu, Wyckliff P. Omondi, Doris W. Njomo.

**Visualization:** Collins Okoyo, Omar Kopi.

**Writing – original draft:** Collins Okoyo, Omar Kopi, Paul M. Gichuki, Bridget W. Kimani, Doris W. Njomo.

**Writing – review & editing:** Collins Okoyo, Omar Kopi, Paul M. Gichuki, Bridget W. Kimani, Tabitha Kanyui, Titus Waititu, Wyckliff P. Omondi, Doris W. Njomo.

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
