## [Decision Letter · Decision Letter 0]

2 Mar 2025

PNTD-D-24-01891

Assessing the effectiveness of the community participation approaches to improve access to mass drug administration for trachoma elimination in a pastoral conflict area of Baringo County, Kenya

Dear Dr. Okoyo,

Thank you for submitting your manuscript to PLOS Neglected Tropical Diseases. After careful consideration, we feel that it has merit but does not fully meet PLOS Neglected Tropical Diseases's publication criteria as it currently stands. Therefore, we invite you to submit a revised version of the manuscript that addresses the points raised during the review process.

Please submit your revised manuscript within 60 days May 01 2025 11:59PM. If you will need more time than this to complete your revisions, please reply to this message or contact the journal office at plosntds@plos.org. Please include the following items when submitting your revised manuscript:

We look forward to receiving your revised manuscript.

Kind regards,

Mathieu Picardeau

Section Editor

Shaden Kamhawi

co-Editor-in-Chief

Paul Brindley

co-Editor-in-Chief

**Journal Requirements:**

1) Please note that the Author Summary should appear in your manuscript between the Abstract (if applicable) and the Introduction, and should be 150-200 words long. The aim should be to make your findings accessible to a wide audience that includes both scientists and non-scientists. Sample summaries can be found on our website under Submission Guidelines:

3) Please note that the Response to Journal Requirements should not be uploaded as Supporting Information.

**Comments to the Authors:**

**Please note that one of the reviews is uploaded as an attachment.**

**Reviewers' Comments:**

Reviewer's Responses to Questions

**Key Review Criteria Required for Acceptance?**

**Methods**

-Are the objectives of the study clearly articulated with a clear testable hypothesis stated?

-Is the study design appropriate to address the stated objectives?

-Is the population clearly described and appropriate for the hypothesis being tested?

-Is the sample size sufficient to ensure adequate power to address the hypothesis being tested?

-Were correct statistical analysis used to support conclusions?

-Are there concerns about ethical or regulatory requirements being met?

Reviewer #1: The objective of the study is well formulated and the pre and post study design is suitable for this objective.

The study population is well identified, clearly described and the analyzed sample responds to the hypothesis or objective set for the study.

The sample size evaluated for the pre- and post-intervention analysis is sufficient and representative of the study population and to infer the results; However, it is considered necessary to describe in the discussion, in the limitations, the representativeness of people under 18 years of age (who did not participate in the surveys), but who were still subject to the mass drug administration.

With respect to the statistical analysis, in the attached document observations are made, related to measures of central tendency used depending on the distribution of the data, the need to clarify whether the ORs, with their 95% CI and p values, are raw values or adjusted? The need to review the reason why some sociodemographic variables are separated from the model and analyzed separately is also identified.

To facilitate the understanding of the article, the authors are suggested to use the lowest risk category as the reference category for the multivariate models, or at least standardize the lowest or highest risk category in all analyses.

It is considered convenient to describe in the methodology the profile of the people who delivered the MDA and the training process they underwent, especially because a high frequency of adverse reactions to medications is reported.

It is considered necessary that the methodology describe how the mass administration of medications is carried out, the recording of data and the assessment of the coverage achieved.

It is considered appropriate to make explicit the strategy used in the intervention to ensure that nomadic people receive treatment.

Reviewer #2: I'm unclear on the timing of the pre-intervention, intervention, and post-intervention periods. On line 146 you say the intervention was the November 2022 MDA, but in line 166, the pre-intervention survey was done in February 2023 and the post-intervention survey was October 2023.

Please include a brief summary of the MDA history of the chosen area. (To help contextualize some of the results about the number of consecutive MDAs people reported taking.)

Were pre-intervention survey results used to formulate the tested interventions, or were these chosen a priori?

More information about the intervention would be useful in order to contextualize some of the results , especially around the suggested strategies on how MDA should be conducted. For example, participants indicated a preference for early morning distribution, but we don't know the timing of the MDA that was actually done.

Why were participant responses pooled and how was this done?

If possible, consider including the quantitative survey instrument as a supplementary file.

Did the information about toilet coverage come solely from the respondents' interviews, or were these directly observed?

I don't see a definition of MDA access. Is this if people were offered the drug, swallowed the drug, or something else? Please clarify.

**Results**

-Does the analysis presented match the analysis plan?

-Are the results clearly and completely presented?

-Are the figures (Tables, Images) of sufficient quality for clarity?

Reviewer #1: The analysis plan was not reviewed for review; observations on this topic are made based on the described methodology.

In general, the data is well presented, however, recommendations and questions are made to the authors about some aspects and data written in the tables that could be improved.

It is considered necessary to adjust some tables in which the risk/protective factors are reported, so as to explain whether the OR, CI and p values are adjusted or crude.

Reviewer #2: The age range listed in line 220 is 16-101, however in the ethics section, you only mention inclusion of respondents over the age of 18. What is the cause for the discrepancy?

I think you could add two columns to table 3 for the number of participants in each category with access to MDA pre- and post-intervention, respectively.

I would switch the order of Tables 3 and 4, to present socio-demographics of respondents prior to odds ratios, but that's a personal preference.

In Table 4, the pre-intervention drinking water source percents are incorrect; likely a copy paste error.

To the header of table 4, please make it clear that the numbers in brackets are percents.

In line 270, you indicate that participants were asked to identify the number of people they knew who were infected. As infection status would require analysis of ocular material via PCR, I'm wondering in instead you meant "affected"? (Affected how so, though? Experiencing trichiasis?)

In Table 6, I don't understand the placement of "CHWs" in the table.

Please define CHW and CHV at first use. Are these interchangeable? In Fig 2, the top panel has CHV and the bottom has CHW.

Line 325: What is the translation of "wazee wa nyumba kumi", please?

I don't understand the results in lines 369-370: how could a higher percentage of people have taken the drugs during the last MDA than those who had ever taken trachoma drugs?

Line 380: Is this "significant" in a statistical sense, or just large? Significantly higher than pre-intervention?

**Conclusions**

-Are the conclusions supported by the data presented?

-Are the limitations of analysis clearly described?

-Do the authors discuss how these data can be helpful to advance our understanding of the topic under study?

-Is public health relevance addressed?

Reviewer #1: A limitation of the study is adequately described, based on the absence of a control group, but it is suggested to review the relevance of including other limitations associated with the non-participation of children under 18 years of age, who did not participate in the study, but did participate in the mass drug administration.

The topic of study is relevant; Understanding the reasons why coverage goals are not achieved in mass drug administration is essential for trachoma elimination programs, as is evaluating the contribution of interventions to improve coverage.

Reviewer #2: I'm not sure if the conclusions that effective communication and enhanced capacity building of MDA teams influenced access to MDA (line 450) is supported by the data presented. (Maybe this is presented and the connection was just not obvious to me! In which case, please clarify.) In general, the results presented allow you to make conclusions on how socio-economic factors are associated with access to MDA pre- and post-intervention. It's less clear that you are able to make conclusions about the effectiveness of the different tested interventions, as ideally you'd have a control group who did not receive the tested interventions. (And you do mention this in the limitations.)

In line 470, you state that the number of participants aware of MDA increased, but from the data this was not significant. I think the conclusion should be qualified with this information.

From Fig 2, aren't most people informed of MDA through Chief's meetings (pre-intervention) and Market place (post-intervention)? I'm not clear on why the conclusion is then that most people are informed by CHWs. (line 472).

In my opinion, the high number of respondents reporting a negative interaction with a CHW also provides evidence that CHW capacity building and training is paramount.

Please include in the limitations an acknowledgement of biases associated with survey-taking, such as social desirability bias (i.e. were respondents biased toward giving answers they perceived as being "correct"?). Was anything done to address these biases, and how do you think they may have affected the results?

**Editorial and Data Presentation Modifications?**

Reviewer #1: It is recommended that tables have footer explanations, to understand whether the ORs are crude or adjusted.

It is suggested that the authors standardize the reference category for the OR analyses; In some variables it is the lowest risk category and in others the highest risk.

Reviewer #2: Data presentation modification suggestions have been included in the relevant sections above.

**Summary and General Comments**

Reviewer #1: Manuscript review:

Assessing the effectiveness of the community participation approaches to improve access to mass drug administration for trachoma elimination in a pastoral conflict area of Baringo County, Kenya

First of all, thank you very much for the opportunity to review this interesting manuscript resulting from a research study. The article evaluates the impact of community participation in improving access to mass drug administration (MDA) in the fight against trachoma in Baringo, Kenya. Using a pre-post intervention design, the study's results show a notable increase in MDA coverage, from 72.4% to 92.9%, suggesting the effectiveness of the implemented strategies. Additionally, there was an improvement in the community's knowledge about trachoma and MDA, indicating that education plays a key role. Overall, the study supports the usefulness of participatory approaches to enhance treatment coverage and recommends their expansion to advance the elimination of trachoma in the region.

Next, I respectfully wish to make a few observations that I believe could be helpful to the authors in improving the quality of the manuscript.

In the author’s abstract

1. Lines 22 and 23: It is suggested to review the text, as it states that the SAFE strategy was developed as a goal to eliminate trachoma. It is considered that the SAFE strategy is a "means," not a "goal."

2. Line 25: The threshold defined by the WHO for mass antibiotic administration is ≥5%, not 10% as stated in the text.

Introduction

3. Lines 90 to 91: In the introduction, it is stated that trachoma is the leading cause of blindness worldwide, which is incorrect because cataracts hold this position. I believe it is necessary to clarify that trachoma is the leading "infectious" cause of blindness.

4. In the introduction, I suggest describing how the distribution of medicines is planned and carried out, and how MDA coverage is calculated in Kenya. Is a population census conducted first, followed by MDA? Are the census and mass drug administration done simultaneously? Are community censuses used or those from sectors other than health, and is treatment coverage estimated through a survey? I suggest clarifying this in the introduction.

Methodes

5. In the methods section, participants aged 18 and older were selected for the study, but it is clear that mass administration of azithromycin should be done starting at 6 months of age. Were ethical reasons the only factor influencing the inclusion of those over 18 in the study? Does absenteeism or non-participation in previous MDA rounds in children and those under 18 explain the same reasons?

6. In the study design, I understand that the baseline coverage data is from the 2022 MDA round; is that correct? It is suggested to clarify this in the text.

7. I believe it is necessary to clarify some variables in the pre-evaluation (2022) and post-evaluation (2023) that may have significantly influenced the MDA coverage. For example: the number of people and groups distributing the medication in both rounds; the amount of time that drug distributors spent in each community conducting the MDA; whether individuals from the community (other than the health sector) participated in the first round of drug administration? If so, how was their MDA training process carried out?

8. It is suggested to clarify in the methodology what the duration of the MDA was in both pre- and post-intervention, how many health workers or drug distributors participated in the pre- and post-intervention; would the difference in number and the time spent in villages explain the increase in coverage?

9. Regarding the time between taking the medication and asking about possible adverse effects in the pre- and post-intervention phases, was the duration similar in both phases? This question considers that not all adverse events occur within the first hours.

10. Lines 165-167: Did it only take 4 days to carry out the MDA in the pre-intervention and 4 days in the post-intervention across the entire district? Please clarify the duration of the pre- and post-MDA rounds in the text.

Results

11. In the results, the average age of the participants is reported. Typically, the age of populations does not follow a normal distribution, and therefore, the mean and standard deviation are not the most appropriate measures of central tendency. Was any test performed to determine the age distribution of the population? What was the result? If the distribution is not normal (which would generally be expected), it is suggested to use the median and interquartile range; please describe this in the methodology.

12. Lines 137 and 138: It is mentioned, "Quantitative data collection methods were used in both the pre- and post-intervention phases." What do you mean by "quantitative methods for data collection"?

13. In Table 3, it is recommended to adjust the decimal places of the percentages, as they total 100.1%.

14. What methodology was used to calculate the impact described in Table 5 (the last column on the right)?

15. In Tables 3 and 5, are the p-values, CI, and OR in the pre-intervention, post-intervention, and impact phases adjusted or crude?•

16. Line 262: In the discussion, I also suggest including a hypothesis that could explain this result: “Participants with floors made of wood/palm/bamboo also showed significantly increased access to MDA (OR = 7.46, 95% CI: 1.01-55.01, p = 0.049)”.

17. In Table 5, to facilitate the interpretation of the results, it is suggested to use the lowest-risk category as a reference, and to do so in a standardized manner for all variable categories. For example, “Never attended school” was used as the reference category, while in other categories, the reference used was the one expected to represent the lowest risk, such as “Post-secondary.” Is there any theoretical basis for this? Otherwise, it is suggested to run the models using the lowest-risk category (or the highest risk category if the focus is on highlighting protective factors), which will make the results easier to understand.

18. Tables 3 and 5 evaluate the relationship between sociodemographic factors and MDA coverage in the pre- and post-intervention phases; the reason for separating the tables and calculating the association measures (OR) separately is unclear. Does this mean that the ORs, confidence intervals, and p-values are from a bivariate analysis? It is suggested to clarify this in the table and methodology, and to perform a multivariable analysis to identify and exclude confounding variables, thus improving the study's external validity.

19. In Table 7, the following data is reported: "Proportion who affirmed to have taken trachoma drugs" (79.4) and 276 (78.9). Wouldn’t these represent the MDA coverage achieved in the pre- and post-intervention?

20. It is unclear in the post-intervention design what methodology or strategy was implemented to address the nomadic population described in the context.

Discussion

21. Table 3 suggests that practicing a religion other than Christianity or not practicing any religion is a protective factor or predictor of better MDA coverage, compared to Christianity. Is there any hypothesis regarding this result? It is considered appropriate to mention this in the discussion.

21. The frequency of the adverse effects described in Lines 342 to 345 is quite significant. It is considered appropriate to mention these values in comparison to those reported by other researchers, to understand whether this is something that is negatively impacting the participation of people in that district in MDA rounds.

Reviewer #2: Thank you for allowing me to review this manuscript. The study done is important for evaluating access to MDA in a particularly challenging region of Kenya. As the trachoma program nears its endgame, these areas will only increase in relevance to the global program, so an understanding of these areas is key! This work is significant for that purpose.

The strengths and weaknesses are well-noted within the manuscript. In my opinion, the analysis of factors associated with MDA access pre- and post-intervention is clear. However, the analysis of the tested interventions is less so. As the authors mention, the lack of a control group here is problematic for drawing conclusions, so I'd advise the authors to be careful in their interpretation of these data.

In addition, since there's no specific space to input comments on the abstract and introduction, please see below:

In the abstract, please rephrase line 72 to avoid the distinction of being a housewife as not being a "meaningful job". (Perhaps it may be more correct to say a job outside the home or a wage-earning job?)

In the introduction, line 90, please edit to reflect that trachoma is the leading infectious cause of blindness (not leading cause of blindness generally). In line 99, is the recommendation based on 10% in the whole population, or children 1-9? Please provide a citation here.

PLOS authors have the option to publish the peer review history of their article (what does this mean? ). If published, this will include your full peer review and any attached files.

**Do you want your identity to be public for this peer review?** For information about this choice, including consent withdrawal, please see our Privacy Policy .

Reviewer #1: **Yes: ** Julián Trujillo Trujillo

Reviewer #2: No

**Figure resubmission:**
---

## [Decision Letter · Decision Letter 1]

11 Jun 2025

PNTD-D-24-01891R1Assessing the effectiveness of the community participation approaches to improve access to mass drug administration for trachoma elimination in a pastoral conflict area of Baringo County, KenyaPLOS Neglected Tropical Diseases  Dear Dr. Okoyo, Thank you for submitting your manuscript to PLOS Neglected Tropical Diseases. After careful consideration, we feel that it has merit but does not fully meet PLOS Neglected Tropical Diseases's publication criteria as it currently stands. Therefore, we invite you to submit a revised version of the manuscript that addresses the points raised during the review process. Please submit your revised manuscript within 30 days Jul 11 2025 11:59PM. If you will need more time than this to complete your revisions, please reply to this message or contact the journal office at plosntds@plos.org. Please include the following items when submitting your revised manuscript: * A rebuttal letter that responds to each point raised by the editor and reviewer(s). You should upload this letter as a separate file labeled 'Response to Reviewers '. This file does not need to include responses to any formatting updates and technical items listed in the 'Journal Requirements' section below. * A marked-up copy of your manuscript that highlights changes made to the original version. You should upload this as a separate file labeled 'Revised Manuscript with Track Changes '. * An unmarked version of your revised paper without tracked changes. You should upload this as a separate file labeled 'Manuscript '. If you would like to make changes to your financial disclosure, competing interests statement, or data availability statement, please make these updates within the submission form at the time of resubmission. Guidelines for resubmitting your figure files are available below the reviewer comments at the end of this letter. We look forward to receiving your revised manuscript. Kind regards, Mathieu PicardeauSection EditorPLOS Neglected Tropical Diseases

Shaden Kamhawi

co-Editor-in-Chief

Paul Brindley

co-Editor-in-Chief

**Reviewers' comments:** Reviewer's Responses to Questions

**Key Review Criteria Required for Acceptance?**

**Methods**

-Are the objectives of the study clearly articulated with a clear testable hypothesis stated?

-Is the study design appropriate to address the stated objectives?

-Is the population clearly described and appropriate for the hypothesis being tested?

-Is the sample size sufficient to ensure adequate power to address the hypothesis being tested?

-Were correct statistical analysis used to support conclusions?

-Are there concerns about ethical or regulatory requirements being met?

Reviewer #1: The objective of the study was “to evaluate the effectiveness of a community-based participatory approach implemented to improve the adoption of MDA for trachoma control”; However, analyses are performed and conclusions are drawn only from univariate/bivariate analyses, meaning the model is not adjusted for potential confounding factors; therefore, the ORs may be under- or overestimated. Therefore, the main recommendation is to perform multivariate analyses to adjust the ORs and their confidence intervals, or, in another scenario, to declare the limitation in the interpretation of the associations presented, given that confounding factors were not controlled.

Reviewer #2: My comments from the previous draft have been appropriately addressed. Many thanks to the authors for this additional effort.

**Results**

-Does the analysis presented match the analysis plan?

-Are the results clearly and completely presented?

-Are the figures (Tables, Images) of sufficient quality for clarity?

Reviewer #1: The results are presented clearly, however, in line with the above, a multivariate analysis is needed.

Reviewer #2: My comments from the previous draft have been appropriately addressed. Many thanks to the authors for this additional effort. Some of the text in the figures is a bit hard to read; however this is likely an issue of the files being compressed as part of the submission system. Please verify the text in the actual figure files is sufficiently clear.

**Conclusions**

-Are the conclusions supported by the data presented?

-Are the limitations of analysis clearly described?

-Do the authors discuss how these data can be helpful to advance our understanding of the topic under study?

-Is public health relevance addressed?

Reviewer #1: The results present high ORs with very wide confidence intervals; although the OR is high (20.12) and the association is statistically significant (p = 0.007), the fact that the CI ranges from 2.29 to 176.96 suggests that the precision of the estimate is low.

This does not invalidate the association, but it does indicate that the result should be interpreted with caution. This limitation should be mentioned in the discussion of the article.

Reviewer #2: My comments from the previous draft have been appropriately addressed. Many thanks to the authors for this additional effort.

**Editorial and Data Presentation Modifications?**

Reviewer #1: Lines 96- 100: The threshold defined by the WHO for mass antibiotic administration is ≥5%, not 10% as stated in the text.

This observation persists in the manuscript; it is widely acknowledged that districts with TF prevalences of 5% to 9.9% should also conduct a round of MDA. A reference is attached. In this same paragraph it is not clear what prevalence is, that is, it is not clear what prevalence of TF is. Reference:

https://pmc.ncbi.nlm.nih.gov/articles/PMC3681669/#:~:text=Background,or%20discontinue%20MDA%2C%20are%20unknown.

Lines 202-206 . Please check: “Although the treatment coverage achieved in 2020 and 2021 was 95.6% and 79.7%, 203 respectively, further investigation at the lower levels (wards) showed coverages ranging 204 between 48% to 57%, which are far below the recommended threshold. Investigations in 205 Loyamorok ward showed low prevalence, ranging from 56.6% and 67.6% respectively in 2020 206 and 2021”.

Specifically on line 205 I think it is necessary to adjust by referring to Coverage and not to prevalence”

Reviewer #2: My comments from the previous draft have been appropriately addressed. Many thanks to the authors for this additional effort.

**Summary and General Comments**

Reviewer #1: Thank you again for the opportunity to review this manuscript. Overall, I believe most of the comments were incorporated or taken into account in the new version, which I consider to be clearer.

My main recommendation is to perform a multivariate analysis to control for confounding factors and avoid over- or underestimation of the ORs used; this would undoubtedly improve the quality of the results and the study in general.

Reviewer #2: My comments from the previous draft have been appropriately addressed. Many thanks to the authors for this additional effort.

PLOS authors have the option to publish the peer review history of their article (what does this mean? ). If published, this will include your full peer review and any attached files.

**Do you want your identity to be public for this peer review?** For information about this choice, including consent withdrawal, please see our Privacy Policy .

Reviewer #1: No

Reviewer #2: No

**Figure resubmission:** While revising your submission, please upload your figure files to the Preflight Analysis and Conversion Engine (PACE) digital diagnostic tool, https://pacev2.apexcovantage.com/. PACE helps ensure that figures meet PLOS requirements. To use PACE, you must first register as a user. Registration is free. Then, login and navigate to the UPLOAD tab, where you will find detailed instructions on how to use the tool. If you encounter any issues or have any questions when using PACE, please email PLOS at figures@plos.org. Please note that Supporting Information files do not need this step. If there are other versions of figure files still present in your submission file inventory at resubmission, please replace them with the PACE-processed versions.**Reproducibility:** To enhance the reproducibility of your results, we recommend that authors of applicable studies deposit laboratory protocols in protocols.io, where a protocol can be assigned its own identifier (DOI) such that it can be cited independently in the future. Additionally, PLOS ONE offers an option to publish peer-reviewed clinical study protocols. Read more information on sharing protocols at https://plos.org/protocols?utm_medium=editorial-email&utm_source=authorletters&utm_campaign=protocols

---

## [Editor Report · Decision Letter 2]

30 Jul 2025

Dear Dr Okoyo,

We are pleased to inform you that your manuscript 'Assessing the effectiveness of the community participation approaches to improve access to mass drug administration for trachoma elimination in a pastoral conflict area of Baringo County, Kenya' has been provisionally accepted for publication in PLOS Neglected Tropical Diseases.

Best regards,

Elsio A Wunder Jr, DVM, Ph.D.

Section Editor

Mathieu Picardeau

Section Editor

Shaden Kamhawi

co-Editor-in-Chief

Paul Brindley

co-Editor-in-Chief

---

## [Editor Report · Acceptance letter]

Dear Dr Okoyo,

We are delighted to inform you that your manuscript, "Assessing the effectiveness of the community participation approaches to improve access to mass drug administration for trachoma elimination in a pastoral conflict area of Baringo County, Kenya," has been formally accepted for publication in PLOS Neglected Tropical Diseases.

Best regards,

Shaden Kamhawi

co-Editor-in-Chief

Paul Brindley

co-Editor-in-Chief
